# Analysis of Obesity among Malaysian University Students: A Combination Study with the Application of Bayesian Structural Equation Modelling and Pearson Correlation

**DOI:** 10.3390/ijerph16030492

**Published:** 2019-02-10

**Authors:** Che Wan Jasimah Wan Mohamed Radzi, Hashem Salarzadeh Jenatabadi, Ayed R. A. Alanzi, Mohd Istajib Mokhtar, Mohd Zufri Mamat, Nor Aishah Abdullah

**Affiliations:** 1Department of Science and Technology Studies, Faculty of Science, University of Malaya, Kuala Lumpur 50603, Malaysia; jasimah@um.edu.my (C.W.J.W.M.R.); ista.ajib@um.edu.my (M.I.M.); mdzufri@um.edu.my (M.Z.M.); aishah.abdullah@um.edu.my (N.A.A.); 2Department of Mathematics, College of Science and Human Studies at Hotat Sudair, Majmaah University, Majmaah 11952, Saudi Arabia; auid1403@hotmail.com or a.alanzi@mu.edu.sa

**Keywords:** obesity modelling, structural equation modelling, Pearson correlation, comparison study

## Abstract

In obesity modelling studies, researchers have been seeking to identify the effective indicators of obesity by using appropriate statistical or mathematical techniques. The main objective of the present study is addressed in three stages. First, a new framework for modelling obesity in university students is introduced. The second stage involves data analysis based on Bayesian Structural Equation Modelling (BSEM) for estimating the Body Mass Index (BMI) (representative of the obesity level) of students at three university levels: Bachelor, Master and PhD. In the third stage, the highest significant correlation is determined between the BMI and other variables in the research model that were found significant through the second phase. The data for this study were collected from students at selected Malaysian universities. The results indicate that unhealthy food intake (fast food and soft drinks), social media use and stress exhibit the highest weightage contributing to overweight and obesity issues for Malaysian university students.

## 1. Introduction

Nowadays, obesity has become a severe worldwide public health problem. Kelly, Yang [1] predicted that by 2030, about 573 million and 1.35 billion adults would have problems with obesity and overweight, respectively. Researchers including Smith [2], Pi-Sunyer [3] and Cameron, Dunstan [4] have published studies concluding that obesity is associated with a higher risk of developing cardiovascular disease, metabolic syndrome and type II diabetes. Some research scholars opine that obesity has become one of the most critical threats to human health over the last two decades [5,6]. Accordingly, focus should particularly be shifted toward obesity prevention efforts, given that public health issues linked to physical conditions are most often consequences of unusual weight gain.

A number of research articles published earlier note and describe indicators of obesity and overweight among university students [7,8,9]. Some researchers have found that BMI (Body Mass Index) is a predictor of good physical function [10,11], sleep quality [12] and smoking habit [13] of university students. However, the majority of such studies that apply to university students lack both model specifications based on student lifestyle and comparative analyses for different levels of education. For instance, some researchers have only employed gender as the primary specification to determine differences in the prevalence of obesity and overweight between female and male university students [14,15,16,17]. The respective studies conclude that female university students exhibit greater concern with controlling their weight, increasing vegetable and fruit intake, paying more attention to exercise, selecting foods lower in sugar and fat, maintaining lower caloric intake and generally being more health-aware.

Students’ socioeconomic status is deemed to be one of the main factors contributing to obesity and overweight according to some studies [13,14,18,19,20]. However, the corresponding research models lack information about the students’ funding sources. Stress is another important factor found to be positively correlated with obesity and overweight in university students [9,21,22]. Stress can be defined as a negative emotional experience accompanied by physiological, behavioural and even biochemical changes [23]. Stress-inducing circumstances not only include university students’ insecurity with their professional and social lives for instance but also factors like insufficient or impaired sleep. In fact, stress has the potential to affect a majority of people emotionally and there is suggestive evidence that stress has a role in developing certain types of depression [24].

In recent years, people of all ages from children to the youth and adults have been relying more and more heavily on digital media devices to perform numerous daily tasks. Intense use of such devices is presumably closely intertwined with daily activities. Many existing studies address physical activity, smoking habit, sleep quality and digital media use separately. However, the authors of the current paper believe that the mentioned activities are all elements encompassed in ‘lifestyle.’ Therefore, based on the structure of latent variables, all abovementioned elements are combined in one latent variable hereby called lifestyle. Other variables examined in this study are individual demographics, mental health and healthy and unhealthy food intake, as well as source of funding as an external variable. Hence, the two independent variables considered in this study are source of funding and demographics, where source of funding includes family support and university students’ income. These two independent variables have not been applied before in this context.

From a mathematical and statistical modelling point of view, descriptive statistics [15,25], ANOVA [26,27] and regression (bivariate or multivariate) are the most familiar techniques used for analysing the associations between various factors and obesity in adults, youth and children. In recent decades, researchers have shown particular interest in obesity modelling with the application of Structural Equation Modelling (SEM) [28]. This methodology facilitates the estimation of BMI (as a dependent variable) based on causal relationships (simple or even complicated) among observed and non-observed (latent) variables. For SEM analysis, several estimators have been introduced in previous studies. Among all estimators, maximum likelihood (ML) is the most commonly used in research related to SEM analysis [29]. However, ML application is often compromised by model misspecification. For instance, models that are too strict with zero residual correlations and exact zero cross-loadings may produce poor model fitting results [30]. Kolenikov [31] and Asparouhov and Muthén [32] demonstrated that the ML estimator presents substantial parameter bias in factor correlations and factor loadings. Due to small sample sizes and normal distribution of independent variables, researchers have started adopting alternative estimators in modelling to overcome the limitations of ML in SEM analysis. So far, only few scholars have suggested applying the Bayesian rather than the ML estimator in SEM analysis to overcome the ML limitations [29,33].

In the literature, there seem to be no comparative studies that examine the patterns influencing university student obesity and overweight using a multilevel framework that involves different education levels. Therefore, the main motivation of this study is to introduce a new framework for estimating obesity among university students and testing the data with enhanced statistical modelling for more accurate results. Considering the advantages and robust prediction power of SEM with Bayesian estimation, the predictor is examined with respect to different levels of education. This approach offers additional knowledge about the predictive power of Bayesian techniques, consequently providing opportunities for future research. In brief, the significant indicator is extracted from SEM analysis and used in Pearson correlation analysis to identify the main association between the research variables and BMI.

## 2. Materials and Methods

### 2.1. Research Framework

Demographics and source of funds are the two main independent variables and BMI level is the dependent variable. Between the independent and dependent variables there are four mediators: lifestyle, healthy food intake, unhealthy food intake and mental health. Figure 1 shows the structure of the research model.

### 2.2. Measurement of Variables

Based on Figure 1, the research framework contains seven variables, namely source of funds, demographics, lifestyle, healthy food intake, unhealthy food intake, mental health and BMI. Source of funds and demographics are defined based on the characteristics of the university students partaking in this study. Lifestyle is measured in terms of health behaviour characteristics introduced by various researchers. The health-related indicators suggested by Nakayama, Yamaguchi [34] and applied by Yanuar, Ibrahim [35] and Ogi, Nakamura [36] as well, are the average working hours, physical activity, smoking habit and average sleep duration. To measure mental health, Boardman [37] and Nakayama, Yamaguchi [34] proposed three main indicators: problems, stress and happiness. In the presents study, healthy and unhealthy food intake are measured based on seven indicators suggested in Kröller and Warschburger [38] research. BMI is a measure of relative size based on an individual’s mass and height [39]. We measured this indicator in line with Escott-Stump [40] study. Appendix A presents more details about measuring the research variables.

### 2.3. Sampling

The cross-sectional study approach is applied in the present work. For cross-sectional type research, relevant data are collected from a population sample at the same point in time. So far, researchers have not been able to deliver normative interpretation nor development substances. Nonetheless, various processes have been suggested for establishing the sample size. Hair, Black [41] recommended structural equation modelling (SEM), which is the most familiar data sampling process. Hair and Black indicated that the sample size should relate to the number of latent variables and the number of indicators to the latent variables. Table 1 shows the sample size selection process based on Hair, Black [41], while Figure 1 presents the six latent variables, one of which (Income) has less than three indicators. Since this comparative study pertains to Bachelor, Master and PhD students, three groups must be considered in the sampling. In order to attain superior output accuracy, at least 300 respondents were required from each of the three education level groups.

## 3. Results

Data collection was carried out at 5 universities in Malaysia. Table 2 describes the proportions of distributed and completed questionnaires. In every university 400 questionnaires were distributed, for an overall total of 2000 (1000—Bachelor, 650—Master and 350—PhD). From the 2000 questionnaires sent, 1773 were completed, which represents a response rate of 88.6%.

### 3.1. Descriptive Statistical Analysis

Table 3 and Appendix A present the distributions of research variables among Bachelor, Master and PhD students. Based on Table 3, 6.1% of Bachelor students sampled (57 of 940) were underweight, 53.4% (502 of 940) in the normal range, 23.0% (216 of 940) overweight and 17.6% (165 of 940) were obese. At the Master level, 10.9% of participants (56 of 512) were underweight, 52.9% (271 of 512) in the normal range, 21.9% (112 of 512) overweight and 14.3% (73 of 512) were obese. In the PhD group, 3.7% of participants (12 of 321) were underweight, 60.1% (193 of 321) in the normal range, 17.8% (57 of 321) overweight and 18.4% (59 of 321) were obese.

### 3.2. BSEM Outputs

#### 3.2.1. Validity and Reliability

In SEM application, Fornell and Larcker [42] suggested some criteria for analysing the validity and reliability of a questionnaire.

Validity: The Cronbach’s alpha index of every latent variable must be evaluated to analyse the validity of the questionnaire. If the Cronbach’s alpha value is higher than 0.7, the validity of the latent variable that is included in some questions is accepted.

Figure 2 denotes the Cronbach’s alpha outputs for the five latent variables among Bachelor, Master and PhD levels. It is notable that all outputs are greater than 0.7. Consequently, the questionnaire validity is accepted based on the research variables.

Reliability: The reliability of the questionnaire is associated with the latent variables and also the indicators of each latent variable. Questionnaire reliability in SEM analysis is recognized depending on two indices: factor loading and average variance extracted (AVE).
Factor loading is evaluated based on every indicator of each latent variable. If a factor loading is higher than 0.7, the intended indicator should be kept with the respective latent variable [43]. Otherwise, the indicator must be eliminated from the latent variable and the remaining SEM data analysis.AVE should be higher than or equal to 0.5 for every latent variable [44].

The factor loading and AVE outputs for the three education levels are presented in Table 4 and Figure 3, respectively. For every group (Bachelor, Master, PhD) some indicators have less than 0.7 factor loading and should therefore be eliminated from the rest of the data analysis. By excluding those indicators, the research reliability is confirmed. Figure 3 illustrates that all latent variables have AVE values greater than 0.5. As a result, the questionnaire and data reliability is accepted.

#### 3.2.2. Test of Normality

Skewness and kurtosis are two common indices introduced by Hair, Black [41] in normality testing with the following conditions:Skewness: the absolute value should be less than 2.Kurtosis: the absolute value should be less than 7.

Table 5 delivers the skewness and kurtosis index values based on the research data gathered for the Bachelor, Master and PhD groups. At this stage, the indicators that were eliminated according to Section 3.2.1 are not tested for normality. Table 5 demonstrates that the normality of all indicators is accepted separately. Furthermore, based on the multivariate normality test output, the kurtosis values are 9.311, 9.021 and 8.887 for bachelor, master and PhD respectively. These values are lower than 10 and in line with Radzi, Jenatabadi [45] suggestion. Thus, multivariate normality is accepted for the data in the present study.

The variables that were eliminated from the normality test are: vegetables, physical activity and work for the Bachelor and Master groups and smoking habit for all groups (Table 5).

#### 3.2.3. Model Fitting

Following reliability, validity and normality analysis, the model fit is tested. Figure 4 illustrates the SEM model fit results. The model fit analysis results are above 0.9 and thus considered acceptable [41], with the goodness of fit index (GFI), normed fit index (NFI), Tucker Lewis index (TLI), comparative fit index (CFI), relative fit index (RFI) and incremental fit index (IFI) measures all within acceptable ranges.

#### 3.2.4. Structural Model

Researchers engaging Bayesian analysis are regularly attempting to infer the priors such that they are informative enough to yield B-SEM advantages [46]. Sensitivity analysis is recommended when there is insecurity regarding prior distribution [47]. In this part of data analysis, the specifications of different priors’ outputs are compared to examine the influence of the priors. To achieve this, the models with four types of prior inputs are compared. Then to assign values to the hyperparameters, a small variance is allocated to each parameter as suggested by Lee [48]. The four prior inputs are calculated accordingly as follows:

Type I Prior: the unknown loadings coefficients are all taken to be 0.5. The values corresponding to $\left\{\beta_{1},\beta_{2},\beta_{3},\beta_{4}\right\}$ are {0.7,0.5, 0.6, 0.7} for the Bachelor, {0.6,0.6,0.5,0.7} Master and {0.7,0.6,0.7,0.5} PhD models. Where,
Hyperparameter β_1_ is the effect of Healthy Food Intake on BMI levelHyperparameter β_2_ is the effect of Mental Health on BMI levelHyperparameter β_3_ is the effect of Lifestyle on BMI levelHyperparameter β_4_ is the effect of Unhealthy Food Intake on BMI levelType II Prior: The hyperparameter values are assessed as half of the values in Prior IType III Prior: The hyperparameter values are assessed as a quarter of the values in Prior IType IV Prior: The hyperparameter values are assessed as double the values in Prior I

The results for the four prior input types are given in Table 6. According to this table, the parameter estimates and standard errors obtained for the various prior types are reasonably close. Therefore, it can be stated that the indices found in terms of BSEM processes are not sensitive to these four prior inputs. As a result, the proposed approach is only valid with the adopted prior and the BSEM applied here is quite robust to the different prior inputs. Accordingly, in discussing the results obtained using BSEM, the results obtained with the type I prior are used.

Based on Table 6, the estimated structural equations that address the relationships among BMI, healthy food intake (*X*_1_), mental health (*X*_2_), lifestyle (*X*_3_) and unhealthy food intake (*X*_4_) for the three levels of education are respectively:
(1)$$\hat{\varphi }\left(Bachelor-BSEM\right)=0.09X_{1}+0.64\;X_{2}+0.67\;X_{3}+\;0.67\;X_{4}$$
(2)$$\hat{\varphi }\left(Master-BSEM\right)=0.11X_{1}+0.31\;X_{2}-0.22X_{3}+\;0.57\;X_{4}$$
(3)$$\hat{\varphi }\left(PhD-BSEM\right)=-0.29X_{1}-0.26\;X_{2}-0.41\;X_{3}+\;0.46\;X_{4}$$

A structural model is used to recognize the hypothesized relationship between research variables, which is linked to the presumed models’ conception. Figure 5, Figure 6 and Figure 7 and Appendix A present the structural Bachelor, Master and PhD models.

The structures of the three models in terms of significance between research variables differ. R^2^ for the PhD model (0.86) is higher than the Master model (0.71) and Bachelor model (0.62). Demographics does not have significant impact on any other variables in the Bachelor model. However, in the Master model demographics has significant impact on lifestyle and mental health, while in the PhD model have significant impact on lifestyle, mental health and healthy food intake.

Figure 8 shows the SEM and regression structures and indicates that in regression modelling, the effect of every independent variable on the dependent variable is based on one direct effect. However, Figure 8 also suggests that the independent variables can affect the dependent variables in SEM in numerous ways. According to the research framework in Figure 8 we have:The effect of ‘Independent1’ on ‘Dependent1’ from direct and indirect effects based on ‘Mediator1’The effect of ‘Independent2’ on ‘Dependent2’ from both indirect effects of ‘Mediator2’ (with a measurement structure) and ‘Mediator3’ (with a latent structure)There is a correlation between ‘Mediator1’ and ‘Mediator3’

The above features can be considered in a single model. However, in regression it is not possible to involve them all in one model.

According to the above statements, it can also be said that SEM is an appropriate approach for obesity modelling. This method has been used in previous studies [38,49] in this area. However, modelling based on regression is the most popular technique employed in a variety of studies in the public health domain. For further verification, a practical analysis was done in this study using four indices to compare regression and SEM, which are representative of the strength and correctness of the prediction analysis. The coefficient of determination (R^2^), mean absolute percentage error (MAPE), root mean square error (RMSE) and mean absolute error (MSE) are the most well-known statistical indices for comparison studies of different modelling methods. Table 7 presents the ordinary least squares (OLS) and BSEM index results for the Bachelor, Master and PhD obesity models.

The R^2^ value for BSEM in all three models is greater than for OLS and the MAPE, RMSE and MSE values for the BSEM outputs are lower than for OLS. Therefore, the performance indices show that BSEM can predict the BMI level better than the OLS model.

### 3.3. Correlation Analysis

The most significant research variables identified based on BSEM analysis are extracted and incorporated for correlation analysis of the three levels of education. The primary objective of this analysis stage is to determine the most significant variables with the greatest effect on the BMI of university students. Table 8 provides the correlation analysis outputs. The highest correlations with BMI for the Bachelor group are for social media use (0.89), fast food (0.86), soft drinks (0.83), stress (0.81) and sweets (0.77). For the Master group, the highest correlations are for fast food (0.93), stress (0.92), soft drinks (0.87), sleep duration (0.84) and social media use (0.83). The highest correlations between BMI and other research variables in the PhD group are stress (0.91), soft drinks (0.82), fast food (0.76), social media use (0.69) and study (063).

## 4. Discussion

This paper represents the first empirical study in Malaysia and elsewhere, in which a multilevel framework is applied to examine educational level differences relative to overweight and obesity. In the first phase of the study a new obesity framework was introduced, which was designed based on university students’ lifestyle. The second phase involved data analysis with focus on the BSEM technique, which fills previous gaps in modelling with the traditional SEM. In the third phase, the most significant research variables with BMI at different educational levels were estimated. The data were collected from 5 universities in Malaysia with the highest numbers of students.

The study results demonstrate that the prevalence of overweight and obesity among Malaysian university students is (21.2%, 16.3%) in the overall study sample, with specific prevalence of (23%, 17.6%) among Bachelor, (21.9%, 14.3%) Master and (17.8%, 18.4%) PhD students, respectively. The general prevalence of overweight and obesity among Malaysian university students is higher than in some other countries [10,14,15,25]. In any case, university presents a time of transition for young adults, which most often involves adapting to a new lifestyle and environment. Pliner and Saunders [50] and Economos, Hildebrandt [51] found that university students generally tend to exhibit some weight gain in the course of their education. Consequently, problems with overweight and obesity potentially arising for university students must be addressed carefully.

The current research framework was designed according to Figure 1. This model is a developed form of Yanuar, Ibrahim [35] model, which was redesigned in the present study keeping in view university students’ lifestyle. Therefore, the fundamental structure of this research model has six latent variables and one observed variable, that is, source of funds, demographics, mental health, lifestyle, healthy food intake, unhealthy food intake and BMI. Source of funds (e.g., income and family support) and demographics (e.g., age and job experience) are the primary independent variables and BMI is the main dependent variable. Four mediators exist between the dependent and independent variables defined, which are lifestyle (including social media use, study time, sleep duration, physical activity, work and smoking habit), healthy food intake (including vegetables, fruits and whole grains), mental health (including happiness, stress, problems) and unhealthy food intake (including sweets, chips, fast food and soft drinks).

For the data analysis based on BSEM in the second research phase, three sets of data for three university education levels (Bachelor, Master and PhD) were employed. In the BSEM process, some indicators were eliminated from the data analysis according to factor loadings. The final BSEM outputs are presented in Figure 5, Figure 6 and Figure 7 for the Bachelor, Master and PhD levels, respectively. The most significant relations are marked as solid black arrows and the non-significant relations are denoted by grey dashed arrows. In terms of the significance or non-significance of the relations, the BSEM outputs differ for the three groups. In the Bachelor model, the first independent latent variable (demographics) has non-significant impact on lifestyle, unhealthy food intake, mental health and healthy food intake. However, the second independent variable (source of funds) has significant impact on all mediators in the research model with different signs. For instance, source of funds has positive and significant impact on lifestyle (0.23), unhealthy food intake (0.51) and mental health (0.31) as well as negative and significant impact on healthy food intake (−0.39). Source of funds has the highest impact on unhealthy food intake and the lowest impact on healthy food intake. In the Bachelor model, lifestyle has significant impact on BMI (0.67) and unhealthy food intake (0.38) and significant impact on mental health and healthy food intake. Mental health has significant impact on BMI (0.64) and unhealthy food intake (0.59) besides non-significant impact on healthy food intake. The coefficient of determination (R^2^) in the Bachelor model is 0.62, meaning that 62% of BMI variation is related to source of funds, demographics, healthy food intake, lifestyle, mental health and unhealthy food intake; the remaining 38% of BMI variation depends on other elements. In the Master model, demographics has significant positive impact on lifestyle (0.37) and mental health (0.31) and non-significant impact on healthy food intake and unhealthy food intake. The same as in the Bachelor model, source of funds in the Master model has significant impact on all mediators: healthy food intake (−0.33), mental health (0.55), unhealthy food intake (0.46) and lifestyle (0.25). In the Master model, source of funds has the most significant impact on mental health among the mediators. The same case applies for the unhealthy food intake mediator in the Bachelor model. The impact of lifestyle on the remaining variables in the Master and Bachelor models differs. In the Master model, lifestyle has non-significant impact on unhealthy food intake, negative and significant impact on BMI (−0.22) and positive and significant impact on both mental health (0.31) and healthy food intake (0.31). The impact of mental health on the other variables (BMI, healthy food intake and unhealthy food intake) in the Master model is the same as the Bachelor model. Mental health has significant impact on both BMI (0.31) and unhealthy food intake (0.33) but non-significant impact on healthy food intake. In the Master model R^2^ is 0.71, which is higher than the Bachelor model. The relationships among research variables in the PhD model completely differ from the Master and Bachelor models. R^2^ in the PhD model is 0.86, which is much higher than the Master and Bachelor models. Demographics has significant impact on all mediators except unhealthy food intake, with the highest impact on healthy food intake (0.53). Source of funds has significant impact on all mediators, as follows: lifestyle (0.37), unhealthy food intake (0.39), mental health (0.62) and unhealthy food intake (0.42). Evidently, source of funds has the highest impact on mental health. In other words, although mindful of healthy food intake, lifestyle, unhealthy food intake and mental health overall, increasing the source of funds is linked to PhD students being more attentive with their mental health (happiness, problems and stress).

Following the data analysis in the second phase with Bayesian structural equation modelling, the data were prepared and the research variables were adjusted for analysis in the third phase. Those research variables that were not significant (shown in Figure 5, Figure 6 and Figure 7) were deleted prior to correlation analysis. Moreover, only data on obesity and overweight were considered for the correlation analysis, mainly because the intention was to identify the variables with the greatest weightage in determining obesity and overweight in participants. Therefore, data for the underweight and normal BMI levels were eliminated from this part of data analysis. The correlation analysis outputs presented in Table 8 illustrate that social media use (0.89) has the highest weightage in obesity and overweight for the Bachelor group participants, followed by fast food (0.86), soft drinks (0.83) and stress (0.81). For the Master students, fast food (0.93) and stress (0.92) have the highest weightages, followed by soft drinks (0.87), sleep duration (0.84) and social media use (0.83). For the PhD level students, stress (0.91), soft drinks (0.85) and fast food (0.76) have the highest weightages, followed by study time (0.63) and social media use (0.2247).

## 5. Conclusions

The application of Bayesian statistics in the health sciences is on the rise. Coinciding with this increased interest, there have been several applications of Bayesian statistics in the field of health analysis over the past few years. The applications of Bayesian statistics include multilevel modelling [52,53], hierarchical modelling [54,55], latent growth modelling [56] and network analysis [57,58]. Within and across each of the studies, researchers have drawn from theory and past empirical work to incorporate weakly informative and informative prior information or employed the default non-informative prior.

This study concerns the obesity risk factor for university students in different levels of education. In this regard, the model was designed based on improvements on previous theories and frameworks and adjusted based on the university student atmosphere. Bayesian SEM and correlation analysis were applied in the present study, which facilitated examining the complexity of university lifestyle as an influence on students’ obesity risk.
In recognizing the complexity of obesity, there is consensus that it is necessary to develop and evaluate a model oriented toward obesity and overweight prevention and treatment. A model-oriented approach can simultaneously address the drivers of obesity at the individual, household, family, community and societal levels through primary and secondary prevention efforts. This study was designed as a model suitable for analysing university student obesity.Bayesian SEM analysis confirmed that the structure risk factor on the BMI level is different for every level of education. This indicates that the level of study not only affects students’ knowledge but it can also affect their perceptions in facing their health environment.For the discussion on student obesity, the associated data records had to be extracted from the entire dataset instead of considering the whole dataset to include the underweight, normal, overweight and obese ranges.

Some limitations were encountered in this study but have led to making a few suggestions for future studies on obesity modelling considering university students as follows:(1)In previous studies, fibre intake [59,60], calorie intake [61,62] and genetics are deemed remarkable indicators of obesity [63,64] and would have been encompassed in our analysis. However, this study has limitations with collecting this type of data and therefore presents a different structure that could not be included in the research model. Nonetheless, it is recommended to analyse these indicators in future studies.(2)The current study is also limited in terms of cross-sectional survey. To provide more confidence in the data analysis accuracy, we suggest running the proposed model with longitudinal data.(3)The outcomes and discussion of this research are restricted by the use of a sample of university students from Malaysia. It is not possible for this sample to be representative of all university students in East Asia. Therefore, the current results do not have sufficient capacity for generalization in other areas. Furthermore, the sample was selected from UM, UPM, USM, UKM and UTM. These universities have high governmental and socioeconomic standards; consequently, selecting different sample structures from other universities may provide a more inclusive picture of university students by taking into consideration religion and political status. We also suggest doing a comparison study of government and private universities.

## Figures and Tables

**Figure 1 ijerph-16-00492-f001:**
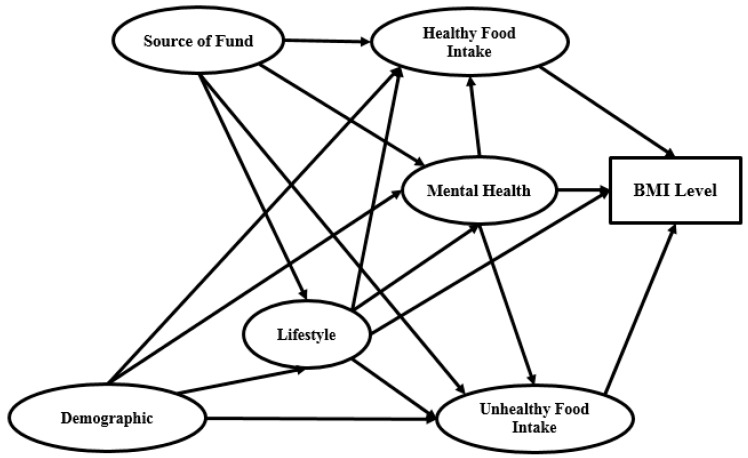
Graphical research model for SEM (Structural Equation Modelling) analysis (BMI: Body Mass Index).

**Figure 2 ijerph-16-00492-f002:**
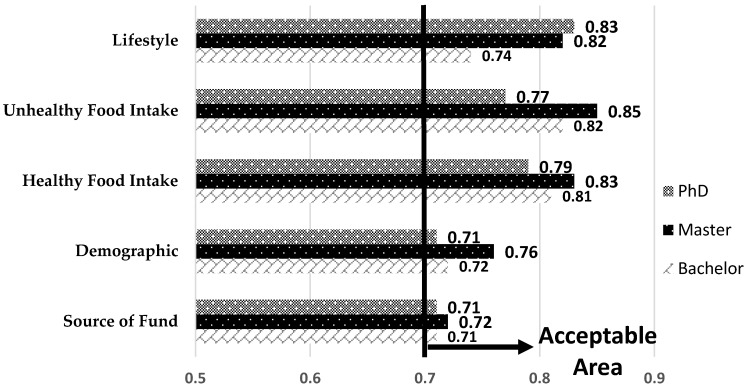
Cronbach’s alpha outputs.

**Figure 3 ijerph-16-00492-f003:**
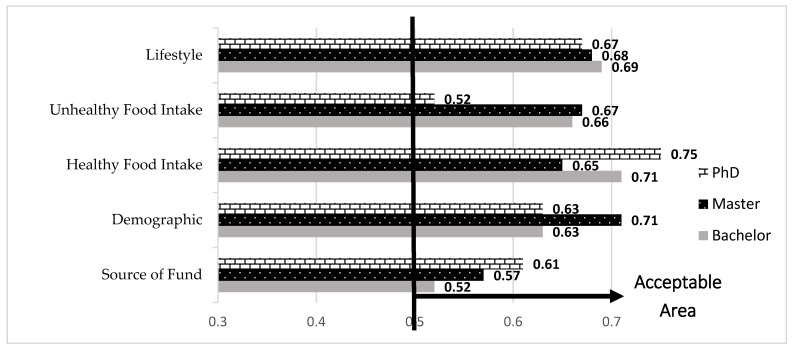
AVE (average variance extracted) analysis outputs.

**Figure 4 ijerph-16-00492-f004:**
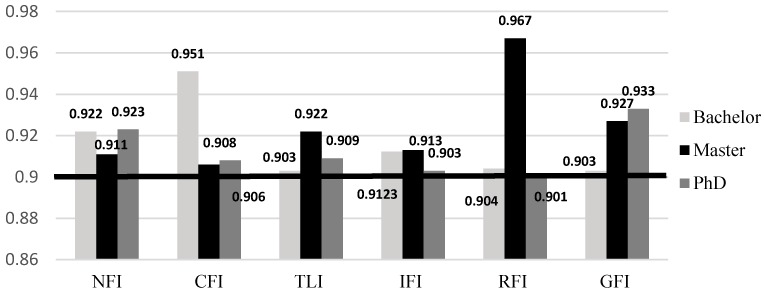
Model fit analysis (NFI: normed fit index, CFI: comparative fit index, TLI: Tucker Lewis index, IFI: incremental fit index, RFI: relative fit index, GFI: goodness of fit index).

**Figure 5 ijerph-16-00492-f005:**
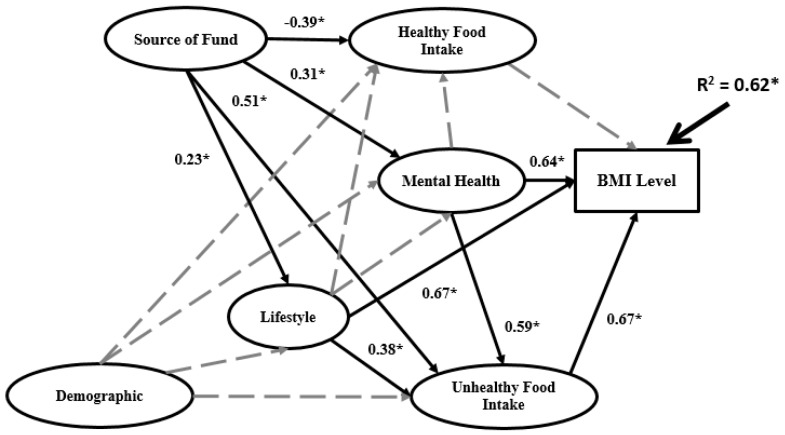
Final BSEM outputs of the Bachelor model.

**Figure 6 ijerph-16-00492-f006:**
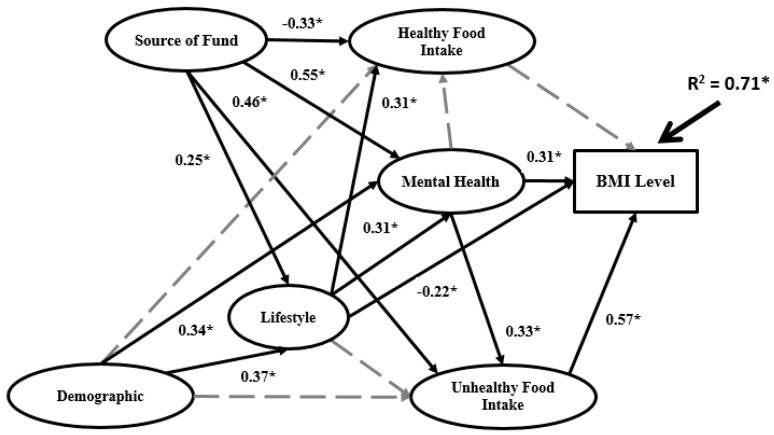
Final BSEM outputs of the Master model.

**Figure 7 ijerph-16-00492-f007:**
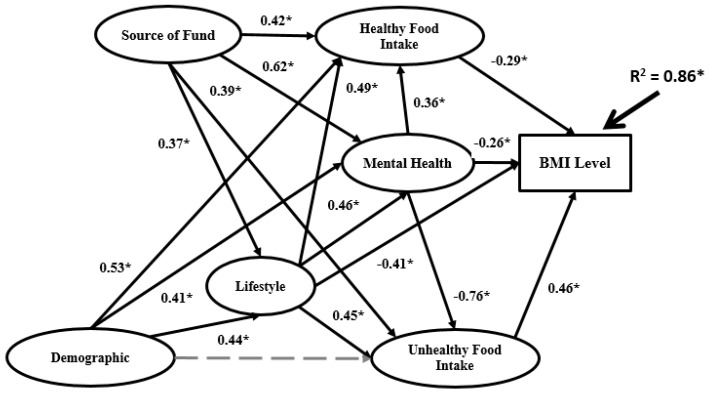
Final BSEM outputs of the PhD model.

**Figure 8 ijerph-16-00492-f008:**
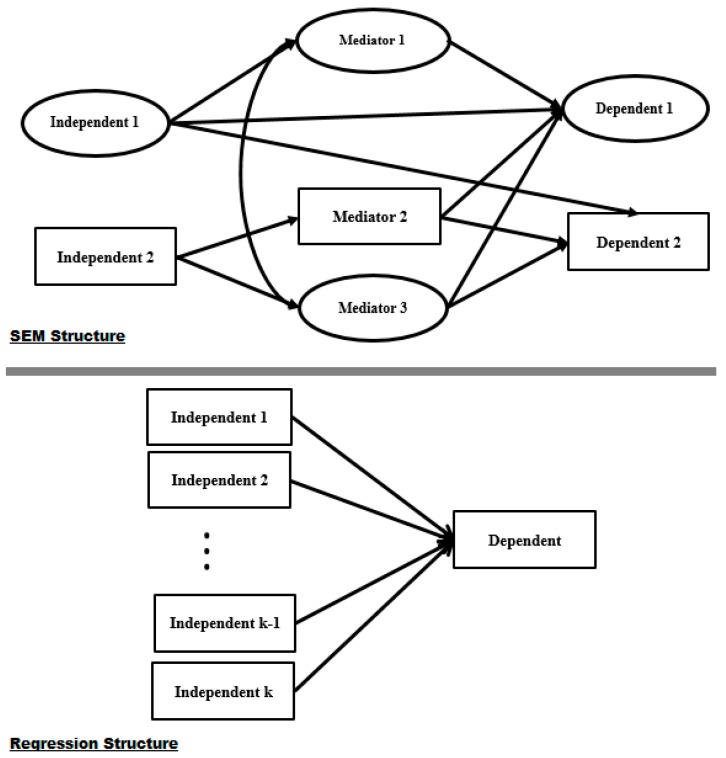
SEM and regression structures.

**Table 1 ijerph-16-00492-t001:** Minimum sample size in SEM (Structural Equation Modelling) according to Hair, Black [41].

Sample Size	Criteria
100 participants	Research model contains 5 or less latent variables and each latent variable covers at least 3 indicators
150 participants	Research model contains 7 or less latent variables and each latent variable covers at least 3 indicators
300 participants	Research model contains 7 or less latent variables and some of the latent variables cover less than 3 indicators
500 participants	Research model contains more than 7 latent variables and some of the latent variables cover less than 3 indicators

**Table 2 ijerph-16-00492-t002:** Sample distribution.

Name of University	Number of Distributed Questionnaires			Number of Completed Questionnaires		
	Bachelor	Master	PhD	Bachelor	Master	PhD
UM	200	130	70	195	122	69
UPM	200	130	70	188	125	61
USM	200	130	70	186	121	65
UKM	200	130	70	192	119	59
UTM	200	130	70	179	125	67
Total	1000	650	350	940	512	321

**Table 3 ijerph-16-00492-t003:** BMI (Body Mass Index) distribution.

Category	Bachelor (Percentage)	Master (Percentage)	PhD (Percentage)	Total (Percentage)
Underweight	6.1%	10.9%	3.7%	7.1%
Normal	53.4%	52.9%	60.1%	54.5%
Overweight	23.0%	21.9%	17.8%	21.7%
Obese	17.6%	14.3%	18.4%	16.8%

**Table 4 ijerph-16-00492-t004:** Output of factor loadings related to latent research variables.

Parameter Description	Bachelor	Master	PhD
Source of Funds			
Income	0.73	0.76	0.77
Family Support	0.78	0.81	0.79
Lifestyle			
Social Media Use	0.88	0.83	0.76
Study Time	0.73	0.81	0.86
Sleep Duration	0.76	0.72	0.74
Physical Activity	0.62	0.66	0.73
Work	0.51	0.55	0.73
Smoking Habit	0.62	0.61	0.68
Demographics			
Education	0.76	0.79	0.71
Age	0.73	0.79	0.71
Job Experience	0.73	0.83	0.86
Unhealthy Food Intake			
Sweets	0.81	0.86	0.87
Chips	0.73	0.72	0.72
Soft Drinks	0.77	0.81	0.71
Fast Food	0.86	0.89	0.81
Mental Health			
Happiness	0.72	0.78	0.84
Problems	0.73	0.81	0.85
Stress	0.76	0.77	0.88
Healthy Food Intake			
Vegetables	0.62	0.68	0.79
Fruits	0.72	0.75	0.88
Whole Grains	0.78	0.81	0.86

**Table 5 ijerph-16-00492-t005:** Normality test based on the most significant indicators according to factor loading analysis.

Indicators	Bachelor		Master		PhD	
	Skew	Kurtosis	Skew	Kurtosis	Skew	Kurtosis
Income	1.26	4.36	0.66	4.17	0.11	3.87
Family Support	0.98	5.32	0.65	1.96	−0.22	−5.09
Vegetables	Removed from data analysis		Removed from data analysis		−1.38	−6.61
Fruits	−0.59	−4.69	1.47	4.19	−1.02	−5.81
Whole Grains	1.11	3.61	0.22	2.76	0.64	3.41
Social Media Use	−0.26	−4.51	0.91	3.91	0.31	2.19
Study Time	1.58	5.67	1.08	5.67	−0.05	−6.96
Sleep Duration	−1.26	−3.08	1.95	3.74	1.11	5.28
Physical Activity	Removed from data analysis		Removed from data analysis		1.27	3.91
Work	Removed from data analysis		Removed from data analysis		1.26	4.51
Smoking Habit	Removed from data analysis		Removed from data analysis		Removed from data analysis	
Age	0.76	3.08	0.28	4.08	0.82	5.66
Job Experience	0.56	4.97	0.33	1.98	−0.77	−2.67
Happiness	0.44	4.08	1.06	4.98	1.07	3.05
Problems	−0.99	−3.78	1.23	5.34	0.47	1.19
Stress	1.78	6.64	1.09	4.19	−0.08	−0.98
Sweets	1.44	5.08	−1.85	−2.98	1.26	2.66
Chips	−0.85	−5.36	1.55	3.91	1.58	5.45
Soft Drinks	−0.46	−1.57	0.91	5.76	−0.88	−2.94
Fast Food	1.08	4.08	−0.44	−4.66	0.27	2.67
BMI	1.77	2.29	1.05	4.96	0.74	4.82

**Table 6 ijerph-16-00492-t006:** Model parameter estimates and standard errors for four types of prior distribution.

Parameter	Type I Prior		Type II Prior		Type III Prior		Type IV Prior	
	Estimate	STD	Estimate	STD	Estimate	STD	Estimate	STD
Bachelor								
β_1_	0.12	0.086	0.09	0.236	0.16	0.096	0.15	0.195
β_2_	0.64	0.055	0.66	0.109	0.61	0.069	0.59	0.096
β_3_	0.67	0.126	0.62	0.177	0.57	0.131	0.68	0.141
β_4_	0.67	0.102	0.54	0.112	0.51	0.129	0.62	0.162
Master								
β_1_	0.10	0.107	0.08	0.111	0.14	0.088	0.16	0.129
β_2_	0.31	0.136	0.36	0.214	0.29	0.151	0.38	0.159
β_3_	−0.22	0.087	-0.18	0.089	-0.29	0.092	-0.21	0.112
β_4_	0.57	0.121	0.55	0.133	0.59	0.131	0.51	0.129
PhD								
β_1_	−0.29	0.063	-0.36	0.093	-0.33	0.111	-0.25	0.098
β_2_	−0.26	0.141	-0.29	0.136	-0.21	0.209	-0.22	0.161
β_3_	−0.41	0.098	-0.39	0.103	-0.32	0.106	-0.40	0.123
β_4_	0.46	0.127	0.41	0.136	0.51	0.133	0.44	0.202

**Table 7 ijerph-16-00492-t007:** Comparison of BSEM (Bayesian Structural Equation Modelling) and OLS index results in different models.

Index.	BSEM (Bachelor)	BSEM (Master)	BSEM (PhD)	OLS (Bachelor)	OLS (Master)	OLS (PhD)
MAPE	0.65	3.59	2.22	2.47	6.91	4.29
RMSE	2.68	0.58	1.57	3.99	2.19	4.89
MSE	1.69	2.36	0.98	5.69	7.39	2.33
R^2^	0.62	0.71	0.86	0.51	0.58	0.61

**Table 8 ijerph-16-00492-t008:** Pearson correlation results for the most significant indicators from the BSEM process.

Indicators	Bachelor	Master	PhD
Income	0.41	0.49	0.55
Family Support	0.55	−0.59	0.07
Vegetables	Deleted (Table 5)	Deleted (Table 5)	−0.34
Fruits	Deleted (Figure 5)	Deleted (Figure 6)	−0.19
Whole Grains	Deleted (Figure 5)	Deleted (Figure 6)	0.13
Social Media Use	0.89	0.83	0.69
Study	−0.59	−0.52	0.63
Sleep Duration	−0.62	0.84	−0.59
Physical Activity	Deleted (Table 5)	Deleted (Table 5)	−0.38
Work	Deleted (Table 5)	Deleted (Table 5)	0.08
Smoking Habit	Deleted (Table 5)	Deleted (Table 5)	Deleted (Figure 7)
Age	Deleted (Figure 5)	0.18	0.48
Job Experience	Deleted (Figure 5)	0.41	0.42
Happiness	−0.71	0.71	0.11
Problems	0.72	−0.63	0.52
Stress	0.81	0.92	0.91
Sweets	0.77	0.75	0.51
Chips	0.74	0.77	0.51
Soft Drinks	0.83	0.87	0.85
Fast Food	0.86	0.93	0.76

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
