# Peer review of "Analysis of Obesity among Malaysian University Students: A Combination Study with the Application of Bayesian Structural Equation Modelling and Pearson Correlation"

_ijerph, 2019, doi:10.3390/ijerph16030492_

Reviewer 1 Report

p1, L38: The expression body situation is not adequate.
P2, L48-49: This sentence should be presented after the literature review.
P2, L54-55: Same as above.
P2, L62-63: This sentence should be part of the first paragraph when authors are pointing out the relevance of obesity. 

P2, L53: It is not clear what the authors meant to say. What does it mean to state that these researches were implemented in individuals? Similar statement is presented in P1, L41. Do the authors mean to say individual level? If yes, please explain the reason why the authors do not believe the analysis using individual level data is enough to study indicators of obesity.
P2, L67: Please remove "were".
P2, L70: Do the authors mean multivariate or multivariable? Please read  https://www.ncbi.nlm.nih.gov/pmc/articles/PMC3518362/
P2, L72: Do the authors explicit the meaning of SEM in their first use?

P2, L76: Please add references to state that ML is the most common estimator used in many studies based on SEM analysis.
P2, L80-82: Please rewrite the sentence: "The disadvantages of SEM analysis ..."
P2, L86: What do the authors mean by stronger technique?  Would it be mode adequate to small sample sizes?

P3, L110-111: What is the currency?
P6, Table 4: Please add the total sample size of each group at the side of category name.

P8, L237: Please cite reference with such guidelines.
P9, L242: Which multivariate normality tests?
P9, L243: What are these numbers between parenthesis?
P9, Table 6: Please create a legend to indicate that the indicator was removed from the data analysis.
P9, L246, Please add details about the model and Bayesian approach: What are the parameters of interest? Which priors were chosen? What are the posteriors?  
P9, L249: Is that correct: Primary and high school models?
P11, Table 7: Same as Table 6.
P11, L257: Is not possible to estimate the correlation using SEM instead of calculating Pearson correlation?

Author Response

Reviewer 1:

Note: All of the comments are amended in the paper with blue font color

Comment 1:

p1, L38: The expression body situation is not adequate.

Answer: It’s changed to body condition. Please see line 37

Comment 2:

P2, L48-49: This sentence should be presented after the literature review.

Answer: It’s done. Please see lines 62-67

Comment 3:

P2, L54-55: Same as above.

Answer: It’s done. Please see lines 62-67

Comment 4:

P2, L62-63: This sentence should be part of the first paragraph when authors are pointing out the relevance of obesity.

Answer: It is amended. Please see lines 38-39.

Comment 5:

P2, L53: It is not clear what the authors meant to say. What does it mean to state that these researches were implemented in individuals? Similar statement is presented in P1, L41. Do the authors mean to say individual level? If yes, please explain the reason why the authors do not believe the analysis using individual level data is enough to study indicators of obesity.

Answer: Sorry that sentences. Individual in statistics sometimes is representing of the sample unit. We have changed it all of them in proper way. Please see line 41-42; 51-52; and 55-56.

Comment 6:

P2, L67: Please remove "were".

Answer: It’s done, please see line 60

Comment 7:

P2, L70: Do the authors mean multivariate or multivariable? Please read  https://www.ncbi.nlm.nih.gov/pmc/articles/PMC3518362/

Answer: Thank you so much for your comment. This is our mistake. Sorry for that. You right, in this paper we are not doing multivariate analysis which is needed moderation analysis. Because there is no moderator variables. So I deleted multivariate from that line. Please see line 80.

Comment 8:

P2, L72: Do the authors explicit the meaning of SEM in their first use?

Answer: It is corrected now. Please see line 71. Thank you.

Comment 9:

P2, L76: Please add references to state that ML is the most common estimator used in many studies based on SEM analysis.

Answer: We It is amended. Please see line 75

Comment 10:

P2, L80-82: Please rewrite the sentence: "The disadvantages of SEM analysis ..."

Answer: It’s done. Please see line 79-82.

Comment 11:

P2, L86: What do the authors mean by stronger technique?  Would it be mode adequate to small sample sizes?

Answer: The statement is corrected. Please see line 85-86

Comment 12:

P3, L110-111: What is the currency?

Answer: The currency is RM. Sorry. The related statement is corrected. Please see lines 116-117

Comment 13:

P6, Table 4: Please add the total sample size of each group at the side of category name.

Answer: It’s amended. Please see Table 4. Thank you.

Comment 14:

P8, L237: Please cite reference with such guidelines.

Answer: you right, the references should be there. Thank you. I have added the references. Please see lines 269 and 272

Comment 15:

P9, L242: Which multivariate normality tests?

Answer: This part of is eliminated from the text, as I mentioned in your comment 7. Please see lines 299-300 

Comment 16:

P9, L243: What are these numbers between parenthesis?

Answer: Based on above comment these values are deleted. It was a mistake to do multivariate normality test.  Please see line 299-300.

Comment 17:

P9, Table 6: Please create a legend to indicate that the indicator was removed from the data analysis.

Answer: It’s done, please see 299-300.

Comment 18:

P9, L246, Please add details about the model and Bayesian approach: What are the parameters of interest? Which priors were chosen? What are the posteriors? 

Answer: I have added some statements related to your comments. Please see lines 230-251 and 310-361

Comment 19:

P9, L249: Is that correct: Primary and high school models?

Answer: Sorry, now it’s corrected. Please see lines 364

Comment 20:

P11, Table 7: Same as Table 6.

Answer: Dear reviewer, this two table have different values for normality test and Pearson correlation analysis. We have changed the Table’s title for better understanding. Now, based on the comments two tables added between table 6 and table 7. Please see Table 7 and Table 10.

Comment 21:

P11, L257: Is not possible to estimate the correlation using SEM instead of calculating Pearson correlation?

Answer: SEM include some latent variables and every latent include some measurement variables. For example lifestyle include social media, sleeping, studying, physical activity, working, and smoking habit. SEM is able to show the impact of lifestyle on BMI level and it’s not able to show the impacts of social media on BMI level. Therefore, in this paper we found the significant effective research variables inside the model with SEM and do Pearson correlation with significant variables which are accepted from SEM process. However, if you suggest it is not necessary I will delete it from our analysis. Thank you. 

Reviewer 2 Report

The paper uses Bayesian Structural Equation Modeling and Pearson Correlation to conduct the Obesity Analysis among Malaysian University Students. Generally, the paper is okay, but some important issues should be explained before possible acceptance.

1.         For the Bayesian structural equation modeling, how to determine the structure and parameter?

2.         How to conduct the model validation and verification?

3.         Some Bayesian-related review papers may be helpful, and could be referred to, such as “Application of Bayesian networks in reliability evaluation”, and “Bayesian Networks in Fault Diagnosis”.

4.         Actually, use Bayesian network, we can conduct similar research? What is the difference?

Author Response

Reviewer 2:

The paper uses Bayesian Structural Equation Modeling and Pearson Correlation to conduct the Obesity Analysis among Malaysian University Students. Generally, the paper is okay, but some important issues should be explained before possible acceptance.

Note: All of the comments are amended in the paper with blue font color

1.         For the Bayesian structural equation modeling, how to determine the structure and parameter?

Answer: we have added some statement regarding your comments. Please see line 230-251

2.         How to conduct the model validation and verification?

Answer: Please see lines 301-361

3.         Some Bayesian-related review papers may be helpful, and could be referred to, such as “Application of Bayesian networks in reliability evaluation”, and “Bayesian Networks in Fault Diagnosis”.

Answer: Thank you for your suggestion. I have considered them in the conclusion section. Please see line 514 with yellow highlighted.

4.         Actually, use Bayesian network, we can conduct similar research? What is the difference?

Answer: similar research? to find out the effectiveness for some part is the same. What is the difference? The Bayesian network is totally nonparametric analysis. BSEM is semi parametric. And in BSEM can see R-square, factor loading, and normality test in their analysis. Based on your comments I have prepared one paragraph in conclusion. Please see the lines 511-517.

Reviewer 3 Report

I would like to thank you for giving me this chance to review this manuscript.

The manuscript addresses the effect of demographic, source of found, and other lifestyle factors on the BMI level among three grades of bachelor, master, and PhD based on cross-sectional data in Malaysia. This study introduces the Bayesian Structural Equation Modeling resulting in a combination of several risk factors for the development of overweight and obesity in college students. So far, this approach is uncommon in public health related research and might be of interest for deriving prevention strategies. However, the paper needs to be more clearly structured and the applied methods should be explained in more detail.

The title of the manuscript investigated effective factors on being overweight or obese in university students through Bayesian SEM and Pearson analysis. The focus might be on this variable as it has the highest effect size in the Bayesian structural equation modelling. However, only beta values are shown in the results section and nothing about the corresponding p-values or confidence intervals is mentioned.

More details about research variables in terms of descriptive statistics of latent variables needed.

It is not clear to the reader how the structural equations with Bayesian estimator have been fitted which are resulting in beta coefficients (presented in Figure 4) and what the corresponding confidence intervals or p-values are.

This might also be important to draw any conclusions on the importance of potential risk factors for the development of overweight and obesity.

More details on the different applied modelling techniques might be given in the introduction and the advantages and disadvantages of these approaches should be discussed more detailed in the discussion.

Do the authors claim that this approach yields new findings? If so, would conventional methodologies (OLS) produce drastically different conclusions? In other words, what warrants this new approach if the story is the same?

Author Response

Reviewer 3:

I would like to thank you for giving me this chance to review this manuscript.

The manuscript addresses the effect of demographic, source of found, and other lifestyle factors on the BMI level among three grades of bachelor, master, and PhD based on cross-sectional data in Malaysia. This study introduces the Bayesian Structural Equation Modeling resulting in a combination of several risk factors for the development of overweight and obesity in college students. So far, this approach is uncommon in public health related research and might be of interest for deriving prevention strategies. However, the paper needs to be more clearly structured and the applied methods should be explained in more detail.

Answer: We thank you for your interest in our work and for helpful comments that will greatly improve the manuscript and we have tried to do our best to respond to the points raised. The comments have brought up some good points and we appreciate the opportunity to clarify our research objectives and results. As indicated below, we have checked all the general and specific comments provided by you and have made necessary changes accordingly to your indications.

Note: The revision based on your comments is typed by blue color font in the paper.

The title of the manuscript investigated effective factors on being overweight or obese in university students through Bayesian SEM and Pearson analysis. The focus might be on this variable as it has the highest effect size in the Bayesian structural equation modelling. However, only beta values are shown in the results section and nothing about the corresponding p-values or confidence intervals is mentioned.

Answer: It’s done. Please see Table 9

More details about research variables in terms of descriptive statistics of latent variables needed.

Answer: It’s done. Please see Table 5

It is not clear to the reader how the structural equations with Bayesian estimator have been fitted which are resulting in beta coefficients (presented in Figure 4) and what the corresponding confidence intervals or p-values are.

Answer: It’s amended. Please see lines 301-308.

This might also be important to draw any conclusions on the importance of potential risk factors for the development of overweight and obesity.

Answer: Please see lines 523-528.

More details on the different applied modelling techniques might be given in the introduction and the advantages and disadvantages of these approaches should be discussed more detailed in the discussion.

Answer: It’s done. Please see lines 479-490

Do the authors claim that this approach yields new findings? If so, would conventional methodologies (OLS) produce drastically different conclusions? In other words, what warrants this new approach if the story is the same?

Answer: It’s done. Please see lines 493-507.

Round  2

Reviewer 1 Report

The language needs extensive corrections. I provided some of them in my comments. The paper is still far from publishable. Not all previous comments were addressed. See below comments to the authors below:
The authors studied the association between obesity and prognostic factors among students using a Structural Equation Model under a Bayesian approach.
- Introduction
L34 Did Rao, and Pandey and Khan just believe or demonstrate that obesity has become one of the most vital human threats over the last two decades? Believe something sounds more as an opinion, which weakens any statement and makes it not adequate for an introduction of a scientific paper.
L38-L39. The added sentence is out of place. It should be move to, for example, L58.
L42. What kind of specifications?
L48-L49. "In previous studies, ..." this sentence should be in the last paragraph when the authors introduce the novelty of their paper as previously suggested.
L52-L58. Please rewrite this sequence of sentences as one or two informative sentences. "Stress was another...", "Stress of the student...", "Stress can be ...", "Stress can have.."
L58-L59. "Some researchers found that BMI..." this sentence should be part of the first paragraph when the authors highlight the relevance to study overweight among students as previously suggested.
L59. "Most of the previous..." is a new paragraph.
L69. Please use plural: "Descriptive statistics[11, 25], ANOVA [26, 27], regression are the most common techniques". In addition, these methods are not used to study obesity. They are used to study the association between prognostic factors and obesity.
L78. Do Kolenikov, Asparouhov and Muthén believe or demonstrated that the ML estimator has extensive bias? Same as L34.
L80. Correction: presence instead of present.
- Materials and Methods
L97 Correction: structure of the research model.
L108 - L174 The questionnaire should be a supplemental material instead of a series of bullet points. The current presentation is extremely repetitive. The authors should keep and detail the references explaining their measurements.
L116. What does RM stand for?
L169. There is a lack of details about Healthy and Unhealthy food intake.
- Results
L223, Table 5: Number and percentages should be in the same column similar to Table 4. Furthermore, Table 4 and 5 would benefit if the format number (percentage) is used instead of number; percentage.
- Test of Normality
The normality multivariate test could be used because the authors are following Jenatabadi et al (2017), which states that Y follows a Normal multivariate distribution.
L290-L291. Please present references to justify these criteria for normality as previously suggested. The references added justify reliability criteria.
2
L299. Correction: vegetables.
- BSEM Outputs
This section should be under methodology. It is need more details to be understandable. Jenatabadi et al (2017) presents more details about the methodology, but it is not complete either. Lee (2008) probably presents all the details, but it is not promptly accessible because it is a book. I would suggest to omit this section and refer to an accessible paper in addition to Jenatabadi et al (2017). My suggestion is:
Yanuar F, Ibrahim K, Jemain AA. Bayesian structural equation modeling for the health index. Journal of Applied Statistics. 2013 Jun 1;40(6):1254-69.
Notice that this paper assumes the measurement variables as continuous, which is not true for the independent variables, only BMI.
Another option is that the authors should use the above suggested paper to give more details about the Bayesian approach as listed below:
L229. The authors have not defined BSEM before using it.
L230. What is the connection between X and Y?
L232. What is the probability distribution of Y?
L234. What are the equations? I am guessing that the equations are Y = lambda Omega + Epsilon and Omega = Lambda_omega Omega + Delta
L234. What is the interpretation of the parameters? Please clarify. For example, \tau are the thresholds for the latent continuous variables Y as showed in Jenatabadi et al (2017). I am guessing that: \Phi is the covariance matrix of measured variables; \Lambda is the vector of coefficients that regress the latent variables \Omega on Y; \epsilon is the vector of residuals associated with Y; \Lambda_omega is the vector of coefficients of structural equations; \delta is the vector of residuals associated with \Omega.
L250. Which software was used to implement the MCMC methods?
- Model Fitting
L304-305. Please give reference for these goodness of fit measures.
- Structural Model
L310-L311. This sentences is not coherent. Please rewrite it.
L315. Correction: evaluated
L317-L320. What are the values for the hyperparameters for Prior I? Otherwise, it is not possible to understand of the other priors because they are were chosen based on variations of Prior I.
L322-L326. It is not clear what the authors meant to state in this paragraph. Please rewrite. Why did the authors choose different priors for different levels of education?
Figure 5 is not readable. Is possible to add a higher resolution?
Table 9. What does CI stand for? Under a Bayesian approach, investigators calculate Credible Intervals instead of Confidence Intervals. Moreover, there is not clear definition of a Bayesian p-value. If the authors are showing a classical p-value, then it should be removed because it is coherent with their Bayesian approach. If it is a Bayesian definition of p-value is being used, the authors should cite a reference discussing that.
- Discussion
L479-L507 should be presented under Results section.

Author Response

Reviewer 1:

The language needs extensive corrections. I provided some of them in my comments. The paper is still far from publishable. Not all previous comments were addressed. See below comments to the authors below:
The authors studied the association between obesity and prognostic factors among students using a Structural Equation Model under a Bayesian approach.
Dear Reviewer
Thank you for your second comments and nice help regarding the manuscript. After we have done all the comments, the manuscript proofread by a native English expert. Many of our publications especially in MDPI have done proofread by her.
All of your second comments were answered with green font colour in the manuscript. The blue font colour belonged to the first revision.
- Introduction Comment 1
L34 Did Rao, and Pandey and Khan just believe or demonstrate that obesity has become one of the most vital human threats over the last two decades? Believe something sounds more as an opinion, which weakens any statement and makes it not adequate for an introduction of a scientific paper.
Answer: we rewrite the statement based on your comment. Please see lines 34-35.
Comment 2
L38-L39. The added sentence is out of place. It should be move to, for example, L58.
Answer: It’s done, please see lines 58-60
Comment 3
L42. What kind of specifications?
Answer: I have added some words. Please see line 41-42
Comment 4
L48-L49. "In previous studies, ..." this sentence should be in the last paragraph when the authors introduce the novelty of their paper as previously suggested.
Answer: It’s amended. Please see lines 85-87
Comment 5
L52-L58. Please rewrite this sequence of sentences as one or two informative sentences. "Stress was another...", "Stress of the student...", "Stress can be ...", "Stress can have.."
Answer: it is done. Please see lines 51-57
Comment 6
L58-L59. "Some researchers found that BMI..." this sentence should be part of the first paragraph when the authors highlight the relevance to study overweight among students as previously suggested.
Answer: It is done. Please see lines 39-40.
Comment 7
L59. "Most of the previous..." is a new paragraph.
Answer: It is done. This comment is related to your comment 2. Please see line 58.
Comment 8
L69. Please use plural: "Descriptive statistics[11, 25], ANOVA [26, 27], regression are the most common techniques". Answer: it is done. Please see line 70. In addition, these methods are not used to study obesity. They are used to study the association between prognostic factors and obesity. Answer: we revised the statement based on your comment. Please see line 71
Comment 9
L78. Do Kolenikov, Asparouhov and Muthén believe or demonstrated that the ML estimator has extensive bias? Answer: it is corrected based on your comment. Please see line 80 Same as L34. Answer: This comment is related to comment 1 and the statement is totally changed. Please see lines 34-35.
Comment 10
L80. Correction: presence instead of present.
Answer: The statement related to this correction totally is changed. Please see lines 81-83.
- Materials and Methods Comment 11
L97 Correction: structure of the research model.
Answer: it is done. Please see line 100.
Comment 12
L108 - L174 The questionnaire should be a supplemental material instead of a series of bullet points. The current presentation is extremely repetitive. The authors should keep and detail the references explaining their measurements.
Answer: it is done. Please see lines 110-161.
Comment 13
L116. What does RM stand for?
Answer: I put the information which is RM representative of Ringgit Malaysia. Please see line 118.
Comment 14
L169. There is a lack of details about Healthy and Unhealthy food intake.
Answer: it is done. Please see lines 159-161.
- Results Comment 15
L223, Table 5: Number and percentages should be in the same column similar to Table 4. Furthermore, Table 4 and 5 would benefit if the format number (percentage) is used instead of number; percentage.
Answer: Table 4 and Table 5 are changed based on your comments.
Comment 16
- Test of Normality
The normality multivariate test could be used because the authors are following Jenatabadi et al (2017), which states that Y follows a Normal multivariate distribution.
Answer: it is added with different way. Please see lines 258-261.
Comment 17
L290-L291. Please present references to justify these criteria for normality as previously suggested. The references added justify reliability criteria.
Answer: Thank you for your comment. The reference is added. Please see line 251.
Comment 18
L299. Correction: vegetables.
Answer: I am so sorry for this mistake and thank you for your help. It’s corrected. Please see line 265.
Comment 19
- BSEM Outputs
This section should be under methodology. It is need more details to be understandable. Jenatabadi et al (2017) presents more details about the methodology, but it is not complete either. Lee (2008) probably presents all the details, but it is not promptly accessible because it is a book. I would suggest to omit this section and refer to an accessible paper in addition to Jenatabadi et al (2017).
Answer: With your comment those parts are omitted from the manuscript. Therefore, the following comments (highlighted words) didn’t amend for the revision. My suggestion is: Yanuar F, Ibrahim K, Jemain AA. Bayesian structural equation modeling for the health index. Journal of Applied Statistics. 2013 Jun 1;40(6):1254-69. Notice that this paper assumes the measurement variables as continuous, which is not true for the independent variables, only BMI. Another option is that the authors should use the above suggested paper to give more details about the Bayesian approach as listed below: L229. The authors have not defined BSEM before using it. L230. What is the connection between X and Y? L232. What is the probability distribution of Y? L234. What are the equations? I am guessing that the equations are Y = lambda Omega + Epsilon and Omega = Lambda_omega Omega + Delta L234. What is the interpretation of the parameters? Please clarify. For example, \tau are the thresholds for the latent continuous variables Y as showed in Jenatabadi et al (2017). I am guessing that: \Phi is the covariance matrix of measured variables; \Lambda is the vector of coefficients that regress the latent variables \Omega on Y; \epsilon is the vector of residuals associated with Y;
\Lambda_omega is the vector of coefficients of structural equations; \delta is the vector of residuals associated with \Omega. L250. Which software was used to implement the MCMC methods?
- Model Fitting Comment 20
L304-305. Please give reference for these goodness of fit measures.
Answer: It is done. Please see line 270.
- Structural Model Comment 21
L310-L311. This sentences is not coherent. Please rewrite it.
Answer: It is done. Please see line 276-277.
Comment 22
L315. Correction: evaluated
Answer: It is done. Please see line 281.
Comment 23
L317-L320. What are the values for the hyperparameters for Prior I? Otherwise, it is not possible to understand of the other priors because they are were chosen based on variations of Prior I.
Answer: You right, and it is amended. Please see line 283-285.
Comment 23
L322-L326. It is not clear what the authors meant to state in this paragraph. Please rewrite. Answer: It is done. Please see lines 289-294. Why did the authors choose different priors for different levels of education? Answer: Sorry. This was mistaken for data entry for Table 8. All of them are belong to Type I prior. Please see Table 8.
Comment 24
Figure 5 is not readable. Is possible to add a higher resolution?
Answer: It is done. Please Figures 5-7
Comment 25
Table 9. What does CI stand for? Under a Bayesian approach, investigators calculate Credible Intervals instead of Confidence Intervals. Answer: It is done. Please see Table 9 Moreover, there is not clear definition of a Bayesian p-value. If the authors are showing a classical p-value, then it should be removed because it is coherent with their Bayesian approach. If it is a Bayesian definition of p-value is being used, the authors should cite a reference discussing that. Answer: It is removed from Table. Please see Table 9.
- Discussion Comment 26
L479-L507 should be presented under Results section.
Answer: It is done. Please see lines 352-379.

Reviewer 2 Report

I am satisfied with the revision. 

Reviewer 3 Report

I can see the authors

improve the last version of the article and I'm happy with current version.

I recommend to accept the revision.

Round  3

Reviewer 1 Report

Minor comments:

- Table 5 and 9 are essential for the understanding of the reader. It should be model to the supplemental material. The details of the
questionnaire under section 2.2 should be moved to the Supplemental Material as well.

L250: Add space between "Hair, Black [41]" and "in normality";

L278-L280: This sentence is not clear. Please rewrite it.
L282-L284: What are these numbers? Are they the values for the hyperparameters? I guess it will be clear after the previous sentence is rewritten.

- Section 3.3 seems not needed after a more sophisticated analysis has been presented throughout the paper. I would advise focusing on BSEM, but it is the author's decision.

Author Response

Reviewer 1:

Thank you for your second comments and nice help regarding the manuscript. Please note that all of the comments are amended in the paper with blue font color. Comment 1
Table 5 and 9 are essential for the understanding of the reader. It should be model to the supplemental material. The details of the questionnaire under section 2.2 should be moved to the Supplemental Material as well.
Answer: It’s done. Please see three supplemental material files (three attachment files Text S1, Table S1, Table S2) and lines 108-119. Comment 2
L250: Add space between "Hair, Black [41]" and "in normality";
Answer: It’s done. Please see line 178 Comment 3
L278-L280: This sentence is not clear. Please rewrite it.
Answer: It’s done. Please see line 203-206 Comment 4
L282-L284: What are these numbers? Are they the values for the hyperparameters? I guess it will be clear after the previous sentence is rewritten.
Answer: It’s done. Please see line 211-214 Comment 5
- Section 3.3 seems not needed after a more sophisticated analysis has been presented throughout the paper. I would advise focusing on BSEM, but it is the author's decision.
Answer: Dear reviewer, thank you so much for this comment. We would be glad if you allow us to keep correlation analysis which is helping us to find out that “unhealthy food intake (fast food and soft drinks), social media use and stress exhibit the highest weightage contributing to overweight and obesity issues for Malaysian university students” (Please see in our abstract lines 23-25). With only of using BSEM we are not able to find out above information. That is why we want to keep correlation analysis in this paper. Thank you